# Machine Learning for Water Quality Assessment Based on Macrophyte Presence

**Ivana Krtolica** [1,*] **, Dragan Savić** [2,3] **, Bojana Bajić** [1,4] **and Snežana Radulović** [5]

1 The Institute for Artificial Intelligence Research and Development of Serbia, Fruškogorska 1, 21000 Novi Sad, Serbia
2 KWR Water Research Institute, Groningenhaven 7, 3433 PE Nieuwegein, The Netherlands
3 Centre for Water Systems, College of Engineering, Mathematics and Physical Sciences, University of Exeter, Exeter EX4 4QF, UK
4 Faculty of Technical Sciences, University of Novi Sad, Trg Dositeja Obradovića 3, 21000 Novi Sad, Serbia
5 Faculty of Sciences, University of Novi Sad, Trg Dositeja Obradovića 3, 21000 Novi Sad, Serbia
* Correspondence: ivana.krtolica@ivi.ac.rs

**Abstract:** The ecological state of the Danube River, as the world's most international river basin, will always be the focus of scientists in the field of ecology and environmental engineering. The concentration of orthophosphate anions in the river is one of the main indicators of the ecological state, i.e., water quality and level of eutrophication. The sedentary nature and ability to survive in river sections, combined with the presence of high levels of orthophosphate anions, make macrophytes an appropriate biological parameter for in situ prediction of in-river monitoring processes. However, a preliminary literature review identified a lack of comprehensive analysis that can enable the prediction of the ecological state of rivers using biological parameters as the input to machine learning (ML) techniques. This work focuses on comparing eight state-of-the-art ML classification models developed for this task. The data were collected at 68 sampling sites on both river sides. The predictive models use macrophyte presence scores as input variables, and classes of the ecological state of the Danube River based on orthophosphate anions, converted into a binary scale, as outputs. The results of the predictive model comparisons show that support vector machines and tree-based models provided the best prediction capabilities. They are also a low-cost and sustainable solution to assess the ecological state of the rivers.

**Keywords:** Danube River ecological state; support vector machines; k-nearest neighbor; decision trees; random forest; extra trees; naïve Bayes; linear discriminant analysis; Gaussian process classifier





## 1. Introduction

The European Commission Water Framework Directive (WFD) aims to restore and maintain a good ecological state of all water bodies in Europe [1]. The assessment of the ecological state and quality prediction of surface waters is an extremely complex task with no standard algorithm developed for it. Therefore, the development of low-cost models for river ecological state prediction is still a challenging task [2].

The assessment of the ecological state of the Danube River, which is the largest river in the European Union (EU) and the most international river basin in the world, needs the well-organized cooperation of 19 nations. Human activities, such as intensive agriculture operations and civil engineering interventions, may cause pollution and river habitat destruction. More than 20% of the entire Danube River Basin is located in the Pannonian lowland region, which is one of the largest agricultural regions in Europe, with cropland covering around 71% of the basin [3]. Therefore, a high amount of orthophosphate anions in those river sections is expected. Despite these facts, the Danube River Basin still shows a high ecological potential [4]. Starting in 2001, the International Commission for the Protection of the Danube River (ICPDR) organized three Joint Danube Surveys (JDS) to collect

reliable and comparable information on all water quality parameters prescribed by the WFD for the assessment of the ecological state of the whole Danube River Basin. During those scientific expeditions, biological, chemical, physico-chemical and hydromorphological parameter values were collected [5,6].

Macrophytes are biological parameters used for ecological state assessment. They are known as eutrophication indicators that may occur even in river sections with a high amount of orthophosphate anions. Macrophyte presence reflects both water quality and hydromorphological river status. Hydrological alterations caused by dams and flow regulation modify the form of river channels, which in turn affects the structure of aquatic macrophyte species in that river section. Due to their sedentary nature and the ability for nutrient uptake from sediment, when combined with other biological parameters, macrophytes provide the best ecological indication performance [2,7].

As biological and chemical water quality parameters exhibit significant non-linear relationships, a number of studies have used artificial intelligence (AI) models for ecological state prediction [2,8,9]. Particularly in the field of ecology and environmental engineering, AI models are recognized as powerful tools for solving non-linear problems [10]. The presence of macrophytes is easy to ascertain even by naked-eye inspection, which makes them a promising parameter for the assessment of the ecological state of the river. However, there is still a lack of systematic studies to assess the efficiency and effectiveness of various AI models that use macrophytes as input variables for predicting the ecological state of surface waters.

The research by Krtolica et al. [2] describes the assessment of the ecological state of the Danube River Basin using scores of macrophyte presence as input variables. They used feed-forward neural networks (FFNN) to predict the ecological state classes verified via orthophosphate anion concentrations. The current work extends that study by developing a low-cost AI model for in situ river state assessment of long rivers, such as the Danube. Eight machine learning (ML) techniques were used to develop and test classification models: k-nearest neighbors, support vector machines (SVM), decision trees (DT), random forest (RF), extra trees (ET), naïve Bayes (NB), linear discriminant analysis (LDA) and Gaussian process classifier (GPC). All of the developed ML models used the same data set collected during the third Joint Danube Survey expedition (JDS 3), so the results of those modeling approaches are comparable. It is hoped that the comparison of the prediction accuracy of the eight ML models will provide a unique guide for future river ecological state monitoring.

The remainder of the article is organized as follows. Section 2 provides the theoretical background of ML models suitable for predicting the ecological state of surface water. Section 3 presents the research methodology that is used as well as the study area. Section 4 presents the results of the applied models. Section 5 discusses the obtained results, provides conclusions and summarizes the paper's contributions.

## 2. Background

This section describes eight ML techniques suitable for small datasets that were selected for comparison. They will be used to predict the Danube River ecological state classes based on biological variables (i.e., macrophytes) as inputs and classes of ecological state calculated among orthophosphate anion concentrations as outputs.

### 2.1. K-Nearest Neighbour (kNN)

The k-NN is commonly used to predict the class of a new sample point based on datasets in which the data points are separated into several classes [8]. The class of the new sample point is determined based on the distance from its k-nearest neighbors. Even though the kNN methodology is simple, it provides an effective classification method for small datasets. It is, however, inefficient for large datasets because the distance between the new sample point and each other point in the dataset has to be calculated every time the algorithms have to perform another classification. Successful application of the kNN

classification requires an appropriate value to be selected for its parameters, i.e., k, the number of neighbors, and the distance function [9].

### 2.2. Support Vector Machines (SVM)

SVM is a kernel-based learning algorithm that consists of sets of related methods for supervised learning that are applicable to both classification and regression problems [10]. An SVM classifier creates a maximum-margin hyperplane that lies in a transformed input space and splits the example classes while maximizing the distance to the nearest split examples [10]. For nonlinearly separable data, SVM has to map the original input data with nonlinear mapping into another high-dimensional feature space where the maximum interval of classification could be solved [11]. In the field of environmental engineering, SVM modeling on data sets with biological and chemical parameters was applied for predicting bio-indicators of aquatic ecosystems when inputs were environmental factors, physico-chemical parameters and hydromorphology parameters, and the outputs were biological communities (fish, algae and macroinvertebrates) [12]. Adequate parameter and kernel function selection is the key challenge in SVM modeling [12–15]. SVM is good for modeling unknown, partially known, and highly nonlinear complex systems [16].

### 2.3. Naïve Bayes (NB)

As opposed to kNN and SVM, the NB approach belongs to the family of probabilistic classifiers. In classifying direct learning, the function that classes posterior $p(y \mid x)$ is a discriminative model. The basic and most limiting assumption is that all variables are conditionally independent, hence its p $(x \mid y = c) = \prod_{i=1}^{D} p(xi \mid y = c)$ [17]. The method calculates the posterior probability of a class from a prior using Bayes' theorem. The advantage of the NB classifier is that it requires a small number of examples to estimate the means and variances of the variables necessary for classification. However, its output depends on the quality of the prior. The type of Bayesian network where continuous variables are sampled from a Gaussian distribution is known as a conditional Gaussian network (CGN) [18], which is used in this paper.

### 2.4. Decision Tree (DT)

A decision tree classifier creates the classification model by building a decision tree [19]. The original dataset is divided into smaller classes using a recursive algorithm based on a test performed on each node of the tree. Because of its easy implementation, DT is a commonly used algorithm in ML. For the purpose of this study, the CART (classification and regression trees) algorithm is used. The CART algorithm is based on binary splitting of the data and uses many single-variable splitting criteria in determining the best split point. Data is stored at every node to determine the best splitting point [20]. The most significant attribute of decision tree classifiers is their ability to change complicated decision-making problems into simple processes and find an understandable and easy-to-interpret solution [21]. However, they are prone to overfitting and can be unstable because small variations in the data might result in a completely different tree being generated.

### 2.5. Random Forest (RF)

The RF methodology is proven to be an efficient discriminative classifier. RF represents a collection of DT classifiers where each tree depends on the values of a random vector sampled independently and with the same distribution for all trees in the forest. The number of trees is directly proportional to the classifier's accuracy. The process continues until reaching a state in which the generalization error converges on values lower than some threshold, e.g., 10% [22,23]. After a large number of trees are generated, they vote for the most popular class. Combining trees grown using random features can produce improved accuracy [24]. Parameters that describe the selected RF model are the number of estimators, the maximum depth of the tree, the minimum number of samples, the number

of leaves and the function to measure the quality of a split. Verbose and class weight are also significant parameters in all tree-based classifiers.

### 2.6. Extra Trees (ET)

The extremely randomized tree, or extra-tree (ET), algorithm is developed as an extension of the RF algorithm. It can generate a large number of individual DTs from the whole training dataset. The algorithm chooses a split rule from a random subset of features for the root node and a partially random cut point. Selecting a random split parent node divides into two random child nodes. The process repeats with each child node until reaching a leaf node that does not have a child. Finally, all the trees are combined through a majority vote to establish the final prediction. During the construction of the forest, for each feature, the Gini importance needs to be computed. Each feature is ordered in descending order according to its Gini importance [25].

### 2.7. Linear Discriminant Analysis (LDA)

LDA is a commonly used technique for data classification and dimensionality reduction. It works by statistically classifying data into two or more classes using a set of discriminating variables that measure characteristics on which the groups are expected to differ [26]. There are two different approaches to LDA. Data sets can be transformed, and test points can be classified in the transformed space by class-dependent and class-independent transformations. The class-dependent transformation approach involves maximizing the ratio of between-class variance to within-class variance. In class-independent transformation, each class is considered separate from all other classes, and its approach involves maximizing the ratio of overall variance to within-class variance. The LDA approach is quite sensitive to outliers but does not require scaling for successful implementation. Significant parameters that need to be fitted for the LDA modeling process are solver and shrinkage parameters, as well as the number of components and absolute threshold values [27].

### 2.8. Gaussian Process Classifier (GPC)

Gaussian process classifiers provide a probabilistic classification model for datasets [28]. This is another Bayesian nonparametric classification method that calculates posterior probabilities based on a prior. The main advantage of this approach is that it requires no assumptions about the structural form between the input variables and the output. Significant parameters for this kind of modeling are the appropriately chosen kernel function and length scale, the maximum number of iterations, the number of restarts of the optimizer, the warm start parameter and the random state value [29].

## 3. Materials and Methods

### 3.1. Study Area and Field Survey Data

The dataset used for this study included macrophyte species and environmental data extracted from the JDS 3 database [30], which was obtained in the EU-funded SOLUTIONS project. The JDS 3 expedition activities were realized during the summer of 2013. The Danube River Basin was divided into 68 sampling stations, 15 of which were located in the mouths of various tributaries (Figure 1). With the exception of a few inaccessible sampling points, both the left and right river sides were included in the investigation and data collection. There were 123 sampling points in total, where information on the presence of macrophytes and the concentration of orthophosphate anions was collected simultaneously. The abundance assessment of macrophyte vegetation followed European Standard EN 14184, comprising the assessment of individual species and their relative abundance per sampling site [31].

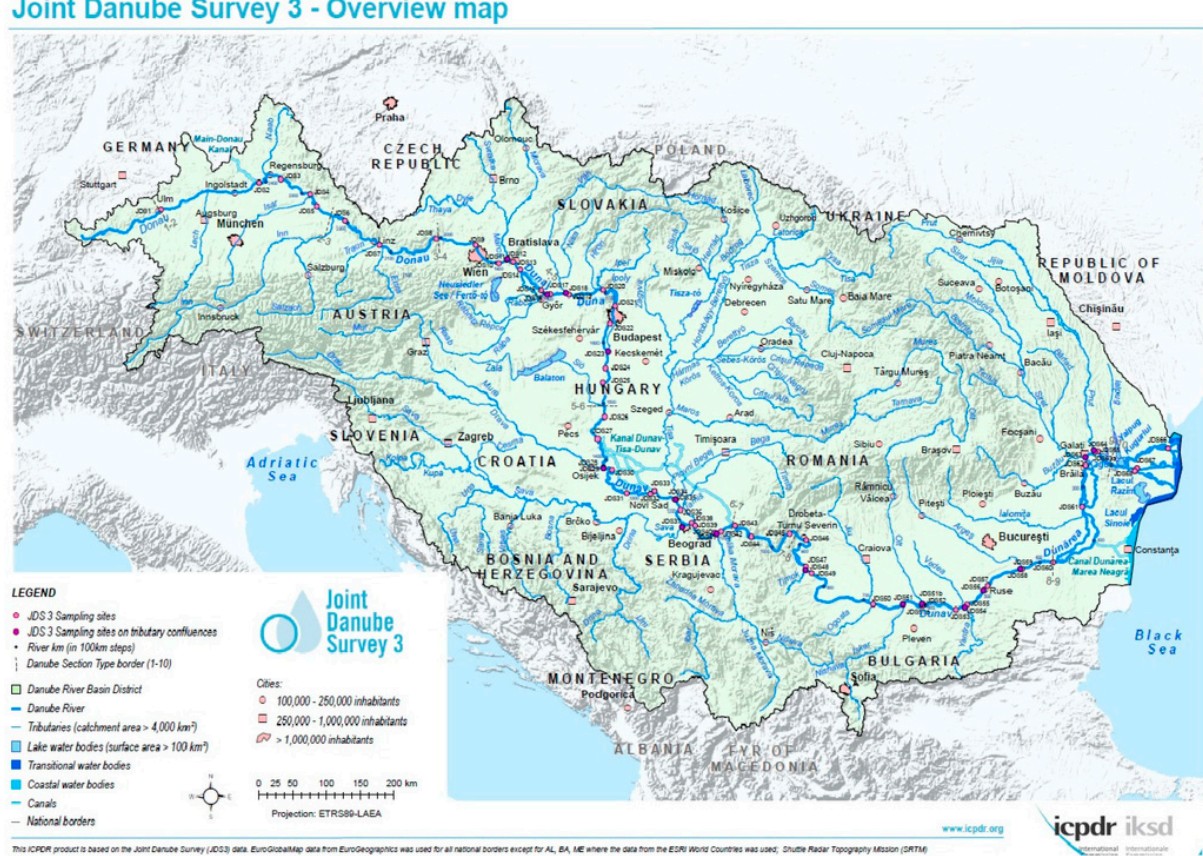

**Figure 1.** JDS 3 overview map. Overview of the JDS3 along the River Danube. Reprinted from the Joint Danube Survey [32].

### 3.2. Data Preparation

The dataset used for modeling included 64 macrophyte types encountered on both sides of the Danube River, which represents 82% of the total taxa found during the JDS 3 expedition. The list of macrophytes used in modeling is available in the paper by Krtolica et al. (2021) [2]. The first step in data preparation was to exclude data with invasive emergent and semi-aquatic macrophytes from the study due to the possibility of them introducing bias to the results.

The second step involved mapping macrophyte data in the input dataset based on the semi-quantitative Kohler method. The method estimates the plant mass index of individual macrophyte species by applying a 5-grade scale [33]. The final macrophyte dataset consisted of 64 plant taxa from 123 sampling sites, where concentrations of orthophosphate anions were also measured.

Due to the lack of a unique classification scale, Krtolica et al. [2] established a new 7-level classification scale for the water quality in the Danube River Basin based on considering national water quality standards in all Danube countries. That classification scale was also used in the modeling described in this paper (Table 1). Those ecological state classes (ESC) based on orthophosphate anion concentration levels were adopted as output (prediction) variables in all ML modeling examples.

**Table 1.** Ecological state classes (ESC) for the Danube River.

| ESC | Orthophosphates (mg/L) | Number of Samples |
|---|---|---|
| I | 0–0.019 | 13 |
| II | 0.02–0.039 | 0 |
| III | 0.04–0.09 | 0 |
| IV | 0.1–0.19 | 96 |
| V | 0.2–0.49 | 12 |
| VI | 0.5–0.8 | 0 |
| VII | >0.8 | 2 |

Lengthwise along the Danube River Basin, four classes of ecological states (I, IV, V and VII) were encountered using the orthophosphate anion concentration data and the scale in Table 1. As class VII is encountered only at one sampling site (JDS 58), on both river sides, this data is omitted from the modeling process due to this being a limited example for the ML methods to learn the input-output relationships. To make a clear distinction between the water that meets the highest quality standard (class I) from more polluted classes, all outputs were converted into values on the binary scale. All river sections that, according to orthophosphate anions, belong to classes IV or V were grouped and assigned a score of 1. All other data points (i.e., those that belong to class I) were assigned a score of 0. All the data points that were assigned a score of 1 are expected to undergo some treatment to address the pollution issues. The two sets of data, i.e., class I (score 0) and class IV + V (score 1), were used as target classes for the ML classifiers.

*3.3. Model Preparation and Evaluation*

All models of ML were developed using Python 3.7 with Keras and TensorFlow deep learning [34]. In all modeling runs, a default set of input parameter values was used. This then allowed a fair comparison of models, as no calibration was performed.

All of the selected ML models were developed to classify the data set (based on the orthophosphate anion presence) from the Danube River according to the macrophyte mass plant index. The selected classifiers are implemented in Python, a powerful interpreter language and a well-founded platform for research. Each model was run 10 times on different subsets of data used for training and validation, with each run initiated using a pseudo-random number generator starting with the same seed. Running a test suite multiple times in random order is performed by using a Python component known as the randomizer. The randomizer first uses a seed to generate deterministic random orders. The output of the randomizer is the result of tests from all of the random orders that were generated [34,35].

The evaluation of experimental results for all models was performed based on the values of the following metrics: (1) classification accuracy, (2) precision, (3) recall, and (4) F1 score.

Accuracy is the proportion of the total number of correct predictions.

$$Accuracy = \frac{Tp + Tn}{Tp + Tn + Fp + Fn}$$

where *Tp* and *Fp* are the numbers of correctly classified outputs for both output classes, and *Tn* and *Fn* are the numbers of misclassified outputs. In this work, the group of samples of class IV + V is considered the positive class (as it refers to a polluted site that needs attention), and the samples of class I are considered the negative class. *Tp (True positive)* is the proportion of correct classifications of the positive class, i.e., *Fp (False positive)* is the incorrect identification of the negative class as positive. *Tn (True negative)* is the measure of the probability that the model will predict the negative class when the true value is

negative. *Fn (False negative)* is the incorrect identification of data as positive when it is, in fact, negative [36–38].

Precision is the proportion of the predicted positive cases that were correct.

$$\text{Precision} = \frac{Tp}{Tp + Fp}$$

Recall or true positive rate (*Tp*) is the proportion of positive cases that were correctly identified.

$$\text{Recall} = \frac{Tp}{Tp + Fn}$$

F1 score is defined as the harmonic mean of precision and recall and presents the most used member of the parametric family of the *F*-measures

$$\text{F1 score} = \frac{2\,Tp}{2\,Tp + Fp + Fn} = 2\,\frac{Precission \cdot Recall}{Precission + Recall}$$

For all four metrics, 0 means the model is performing the worst while 1 means it is performing the best.

### *3.4. Modelling*

A **kNN** algorithm is one of the simplest ML algorithms, also known as "instance-based" learners because they delay data processing until a new instance has to be classified. Because they store data until classification is performed, the classification process is much more computationally intensive than training. This is opposite to the so-called "eager learning" algorithms, such as Bayesian methods, where more time is spent on training. The methodology applied in this work uses the following two second-level tuning parameters and hyperparameters for the kNN model: five neighbors and a leaf size of 30. Uniform weights were used, so all points in each neighborhood were weighted equally. As Euclidean distance is the most widely used in kNN methods, it has been implemented in this case study to assign a class label to the new instance based on the class of the majority of neighbors [39].

**SVM** is one of the state-of-the-art technologies for classification, but the correct model selection is crucial in applying this algorithm [40]. The SVM classification method with the linear support vector for classification (SVC) function was used in this study. The implementation is available in the scikit-learn library, which is appropriate for binary classification. This type of function produces better performance modeling datasets with a large number of samples. The accuracy of SVM depends on the values of its learning regularization metaparameters, which need to be found using an optimization method [41]. A linearly separable set of instances that can be separated by hyperplanes is rarely present in practice, especially in datasets that include chemical measurements. Kernel hyperparameters used for SVM modelling in this study were: the radial basis function (RBF) kernel with the degree-3 polynomial kernel. The size of the kernel cache was chosen to be 200 MB, while the regularization parameter C takes a value of 1.

The **NB** classifier does not use iterative modeling and can work with small datasets. It gives the best results when the features do not correlate with each other because of the NB assumption of class independence, which simplifies the model learning procedure. This study uses a Gaussian NB classifier, which assumes that each class follows a Gaussian distribution. Final modeling results mainly depend on the appropriate choice of hyperparameters [42]. In this study, the following hyperparameters were used for NB modeling: a test size of 0.5 and a random state of 0.

The fact that the **DT** algorithm is non-parametric leads to it efficiently dealing with large and complicated datasets without imposing a complicated parametric structure. A large enough sample size provides a dataset that can be divided into training and validation parts. The main components of a DT model are nodes and branches, and the most important

steps in building a model are the steps of splitting, stopping and pruning [42]. In this study, the following DT hyperparameters were applied: the Gini function to measure the quality of the split, the minimum number of samples to split was two and the minimum number of samples required to be at a leaf node was one. No random state was set, there was no limit imposed on the number of leaves, and all classes are supposed to have a weight of 1. No pruning was performed.

**RF** represents a collection of DT classifiers. The number of trees is directly proportional to the classifier's accuracy. The stopping criterion is when the generalization error converges to values lower than 10% [43]. A decision tree algorithm employs a "greedy" approach that separates the dataset into smaller subsets and takes the simplest solution rather than the most optimal solution. For deciding which feature to split on at each node, the entropy measure (the level of homogeneity of the data subset) needs to be computed. If entropy equals one, then the class labels are equally divided, while an entropy of zero means the sample is completely homogeneous. In the case of a binary classification with only two labels, if the split resulted in the class labels being all 1 (or 0), then the entropy will be zero. The entropy is computed for each variable, and then the difference between the entropy prior to the split and after the split is calculated on each variable. The most useful variable in segmenting the class labels will yield the highest difference in entropies. The hyperparameters used for RF modeling in this study included 100 trees in the forest with no maximum depth of the tree, so nodes were expanded until all leaves are pure or until all leaves contain fewer than the minimum number of sample splits. The minimum number of samples was set to 2, with a minimum of one leaf. Again, the Gini function is used to test the quality of the split. Equal weight classes were used, i.e., weight was set at 1 for both classes, and the verbose parameter was set to 0.

The **ET** algorithm is developed as an extension to the RF methodology with the ability to generate a large number of individual decision trees from the whole training dataset. The algorithm chooses a split rule from a random subset of features for the root node and a partially random cut point. Selecting a random split parent node divides it into two random child nodes, and that process repeats in each child until reaching a leaf node. The final classification is obtained when all the trees are combined through a majority vote. During the construction of the forest, for each feature, the Gini importance is computed, and each feature is ordered in descending order according to it [44]. The same hyperparameters and their default values were used as in the case of RF.

This study uses a **GPC** algorithm with a radial basis function as the kernel of the length scale 1. Only one run is performed, and the maximum number of iterations was set to 100. The warm start was enabled, so the last Newton iteration of the Laplace approximation of the posterior mode is used as initialization for the next call of the posterior mode. A persistent copy of the training data was stored in the object; no random state instance was determined, and no "joblib" parallelism was used at all [45].

The study used the **LDA** solver with singular value decomposition and no shrinkage parameter. The class proportions were inferred from the training data. The number of components was set to a minimum. For the absolute threshold for a singular value of a data sample to be considered significant, which is used to estimate its rank, a value of 0.0001 was used. Because the shrinkage parameter was used, no covariance estimator was used in modeling [46].

## 4. Results

The training process for the selected models involved 2/3 (67%) of the complete dataset. The k-fold cross-validation (CV) method was used to evaluate the accuracy of predictive models. It involves the division of the training dataset into k subsets. During the training, each of the k subsets is used to validate the model, and the data in the other k-1 subsets are aggregated to form a training set. The process is repeated k times, and the model accuracy is evaluated as an average value over all results [47,48]. A 10-fold cross-validation approach (k = 10) is used in the training process in this study. with random

state 1 and the option to shuffle data. Other 1/3 (33%) of the dataset was used for testing, so the model was tested on unknown data. The distribution of the two classes in the training and testing datasets was also kept equal to the distribution in the entire dataset.

Table 2 shows the accuracy results of all classification models with a 10-fold cross-validation approach. In terms of predictive performance, we observed that the overall best models, judged by the accuracy score and standard deviation, were SVM and GPC, which produced the same results.

**Table 2.** Cross-validation training results (averages over 10 training runs).

| Classifier Model | Accuracy | Standard Deviation |
|:---:|:---:|:---:|
| k-NN | 0.88 | 0.056 |
| RF | 0.88 | 0.055 |
| NB | 0.75 | 0.114 |
| SVM | **0.89** | **0.036** |
| LDA | 0.74 | 0.067 |
| DT | 0.87 | 0.091 |
| ET | 0.88 | 0.067 |
| GPC | **0.89** | **0.036** |

Both the training and testing processes used the same architecture. Testing results confirmed the training outcome, with SVM being the most appropriate classifying method for this type of input/output relationship. The testing results of all selected ML classifier methods are shown in Table 3.

**Table 3.** Accuracy, precision and recall rate and F1 score values of selected ML classifier methods.

| Classifier Model | Accuracy | Precision Rate | Recall | F1 Score |
|:---:|:---:|:---:|:---:|:---:|
| k-NN | 0.82 | 0.87 | 0.94 | 0.90 |
| RF | 0.85 | 0.87 | 0.97 | 0.92 |
| NB | 0.62 | 0.86 | 0.69 | 0.76 |
| SVM | **0.88** | **0.88** | **1.00** | **0.93** |
| LDA | 0.62 | 0.86 | 0.69 | 0.76 |
| DT | 0.85 | 0.87 | 0.97 | 0.92 |
| ET | 0.85 | 0.87 | 0.97 | 0.92 |
| GPC | 0.77 | 0.86 | 0.89 | 0.87 |

As can be seen in Table 3, most ML classifiers achieved an accuracy rate above 80%. This is impressive as the models have been developed using the default settings and hyperparameters of the different ML approaches. However, accuracy is not enough to assess how well a classification model predicts data in an imbalanced dataset, such as in this case study. It is, therefore, important that other metrics also achieve high enough values. The use of precision rate, recall and F1 score provides much more in terms of the classifier's performance, thus giving more certainty when all of the metrics indicate good performance. The discrepancy between CV and testing result accuracy is the most significant for the GPC model, which is the consequence of ecological state class distribution.

*Confusion Matrices Based on Testing Results*

Accuracy values of testing models were calculated among modeling results shown in confusion matrices (**4.1.1.–4.1.8.**)

### 4.1.1. kNN

[0    5]

[2    33]

### 4.1.2. RF

[0    5]

[1    34]

### 4.1.3. NB

[1    4]

[11    24]

### 4.1.4. SVM

[0    5]

[0    35]

### 4.1.5. LDA

[1    4]

[11    24]

### 4.1.6. DT

[0    5]

[1    34]

### 4.1.7. ET

[0    5]

[1    34]

### 4.1.8. GPC

[0    5]

[4    31]

The correctly predicted values of the ecological state class are shown on the diagonal, from the top to the bottom right of the matrix. It is interesting to note that most models struggled to predict the class IV + V as there was a small number of those in the training set. This is a serious limitation of the ML methodologies when used for predicting the ecological state class for rivers using a small number of samples.

## 5. Discussion and Conclusions

Ecological state assessment of surface water based on ML is a data-driven approach, which is a typical supervised learning problem [49]. Supervised learning uses labeled input and output data to train ML algorithms to classify new data or predict outcomes correctly. This approach normally relies on a large number of data samples to discover hidden relationships. Most studies dealing with the assessment of the ecological state of a river basin using ML focus on dissolved oxygen in surface water [50]. Furthermore, ML is often used in the context of the prediction of biological parameters based on chemical variables [51].

During each JDS scientific expedition, a relatively small dataset (i.e., 123 samples) was collected due to samples being taken every three kilometers along the Danube [52]. Krtolica et al. [2] used the dataset to develop an ML prediction model for ecological state classes based on the concentrations of dissolved oxygen, nitrates and orthophosphates. Although multilayer feed-forward neural networks are most appropriate for big data samples [53], they managed to achieve an effective model. The question that remains is whether a better ML technique can be found to provide consistently better results and provide further guidance for environmental researchers on how to select the best model.

Eight ML classifying models were developed using the same data set and tested on performing binary classification for the future prediction of river ecological state class based on macrophyte presence and orthophosphate anion concentrations. The methods were used with default settings and hyperparameter values. To provide the best predictive model that can be applied to similar datasets, the performance of all selected ML models was compared using a number of metrics. This work was focused on comparing and evaluating the models in terms of accuracy, precision, recall rate and F1 score. As the F1 score is defined and calculated using the information provided by precision and recall, it

can be considered a single, most informative metric for the comprehensive analysis of the ML modeling process.

Based on the available dataset from the JDS 3 measurement campaign, a two-class classification problem was defined. In this type of classification problem, performance metrics are sensitive to dataset composition, which affects the selection of the best ML algorithm.

Based on all metrics used in this study, SVM shows the best performance among all models. This best performance can be observed in the training results as well as in testing. The success of the SVM can probably be attributed to the non-linearly separable data and the fact that a non-linear kernel function, i.e., RBF, was used in the model. Because the dataset is relatively small, the application of SVM did not require long training times. However, even the best model struggled with correctly predicting the positive class (class IV + V) due to the imbalanced dataset.

In addition to SVM, tree-like methods (RF, DT and ET) have also achieved good results considering all metrics, despite a relatively small dataset. The key reason for that is probably because they are good at handling categorical features in data, which is the case with the JDS 3 data.

The next best performing model is kNN, which has achieved only slightly lower metrics values in the case study. Again, for the default values of the hyperparameter $k$ and the distance measure, this is expected as kNN is robust to noisy training data.

The GPC model is the best among the worst-performing ML models. However, it performed much better (or equally as good as SVM) in training than in testing. Because these models are a generalization of the Gaussian probability distribution and they use a kernel function (similar to SVM), the reason why the GPC model underperformed in comparison with SVM is probably because of data distribution or because of the default values of its hyperparameters.

The worst performing algorithms on this dataset were NB and LDA. Both are, in essence, Bayes classifiers, whereas LDA is not a naive Bayes classifier. In that sense, it is expected that they will perform similarly. The only explanation for the below-par performance is possible violations of model assumptions.

The work in this paper has demonstrated that a few ML models can provide a good prediction model for the ecological state of rivers. Despite a relatively small sample of data, the models have performed well on a number of classification metrics. Training with cross-validation has shown to be good for the imbalanced dataset, but the testing exhibited some weaknesses due to the small number of the particular class (i.e., class IV+V). Generally, an insufficiently large dataset is the main limitation of ML modeling. Thus, applying the selected types of ML models to different (larger) datasets may not result in similar accuracy values. However, for in situ ecological state prediction of surface waters, the ML-based approach can provide information on the eutrophic state of sampling points with high accuracy. This methodology for water quality prediction eliminates the need for costly water sampling and chemical analysis and can serve as a preliminary assessment tool before more complex ecological state estimation.

This work focused on the model-centric approach, where the selection of the appropriate modeling approach is the main goal. Future work will explore the data-centric ML approach, where more focus is on the data, by analyzing, for example, the impact of training data length, temporal resolution, and data uncertainty on forecasting model results. The idea behind the data-centric approach is to improve: (i) the dataset before it is used in a fixed-parameter (out-of-the-box) model and (ii) the level of accuracy that can be achieved.

**Author Contributions:** Conceptualization, I.K. methodology, I.K. and D.S.; software, I.K. and B.B.; validation, I.K., D.S. and B.B.; formal analysis, I.K.; investigation, I.K. and D.S.; resources, S.R.; data curation, I.K.; writing—original draft preparation, I.K.; writing—review and editing, D.S.; visualization, I.K.; supervision, D.S. All authors have read and agreed to the published version of the manuscript.

**Funding:** This research received no external funding.

**Institutional Review Board Statement:** The study was conducted in accordance with the Declaration of Helsinki and approved by the Institutional Review Board.

**Informed Consent Statement:** Not applicable.

**Data Availability Statement:** Not applicable.

**Conflicts of Interest:** The authors declare no conflict of interest.

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
