# Peer review of "Machine Learning for Water Quality Assessment Based on Macrophyte Presence"

_sustainability, doi:10.3390/su15010522_

Round 1

Reviewer 1 Report

The manuscript provided is devoted to an interesting topic, contains important and reliable data collected during the Joint Danube Survey 3, scientific expedition conducted by the International Commission for the Protection of the Danube River (ICPDR) obtained in the EU-funded SOLUTIONS pro- 211 project whereas the expedition activities were realized during the summer of 2013. This is an interesting results that deserves publication once the authors have addressed serious issues.

The title of manuscript „Comparison of machine learning techniques for in situ ecological state prediction of surface waters  is too general. The main achievements  of the research  must be recognizable  „… to predict the Danube River ecological state classes based on biological variables (i.e., macrophytes) as inputs and classes of ecological state calculated among orthophosphate anion concentrations as outputs“.

The Abstract is too extensive (more than 350 words, instead of 200 words maximum), with too extensive background and insufficiently defined clear objectives of the research, as well as the presentation of the obtained research results and the general conclusion.

There is no need to repeat the keywords if they are visible from the title. In the present form the authors list keywords is too broad.

The manuscript sections deviate from the journal guidelines. The authors have introduced a section „ 2. Background“   where they described „ the theoretical background of ML models suitable for predicting the ecological state of surface water“. In the section 3. Materials and methods authors emphasized that „the dataset used for modelling included 64 macrophyte types encountered on both  sides of the Danube River.. “. These results are not presented in the paper. However, in the section “ 5. Discussion and conclusion”   there is no discussion at all, no authors are cited, no discussion is made of the obtained results. The suggestion is to transform part 2 into a section Discussion. And above all, the authors should discuss the obtained results and more precisely defined the conclusion.  

In the “References” there are a number of errors that should be corrected according to the journal's instructions regarding the citation of literature.

Reviewer 2 Report

The manuscript titled as Comparison of machine learning techniques for in situ ecological state prediction of surface waters worked on the selection of the appropriate model among 8 machine learning techniques. The main scientific gap this paper aims is described as a lack of comprehensive analysis that can enable the prediction of the ecological state of rivers using biological parameters as the input to Machine Learning (ML) techniques. This work focused on comparing and evaluating the models in term of accuracy, precision, recall rate, F1 score. The F1 score was defined and calculated by the values of precision and recall. Is F1 score the comprehensive analysis the authors referring to? The authors did not raise a meaningful question to answer nor give a clear answer to it. 

Author Response

Response to the Reviewer #2 comments

Comment #1

This work focused on comparing and evaluating the models in term of accuracy, precision, recall rate, F1 score. The F1 score was defined and calculated by the values of precision and recall. Is F1 score the “comprehensive analysis” the authors referring to?

Authors’ Response:

Thank you for mentioning this. We fulfilled the Chapter Discussion with the explanation that by combining precision and recall values in a single metric, the F1 score, presents an important part of the comprehensive analysis of the Machine Learning (ML) model results that can enable the prediction of the ecological state of rivers using biological parameters as the input to ML techniques. Besides F1 scores, accuracy values for both Cross-validation and testing results also provide significant information about the appropriate choice of ML models (Lines 359 and 360).

Reviewer 3 Report

The authors have presented their case very well. However, there are specific queries I have.

1. How is KNN different from cluster analysis? Can clustering be a tool for ML? How will the results vary for both cases?

2. The equation presented in line 145 is not written correctly. Please modify.

3. In line 227, please mention how you have taken care of redundant data.

4. In line 235, is there any specific reason you have considered for using the classification level proposed by Krtolica et al. [2]?

5. With the limited data that you have, can you justify the accuracy of your results?

There are some typo errors throughout the manuscript which need to be handled.

Author Response

Response to the Reviewer #3 comments

Comment #1

How is KNN different from cluster analysis? Can clustering be a tool for ML? How will the results vary for both cases?

Authors’ Response:

Thank you for your constructive comments.

KNN is a Machine Learning clustering technique, so it is an example of cluster analysis. For KNN calculation of Euclidean distance is important, so it can attempt to cluster/classify the data records close to each other into the same group (Zhao, D., Hu, X., Xiong, S., Tian, J., Xiang, J., Zhou, J., & Li, H. (2021). K-means clustering and kNN classification based on negative databases. Applied Soft Computing, 110, 107732.)

Comment #2

The equation presented in line 145 is not written correctly. Please modify.

Authors’ Response:

Thank you for noticing this important gap. We modified the equation in line 145.

Comment #3

 In line 227, please mention how you have taken care of redundant data.       

Authors’ Response:

The values of orthophosphate anion concentrations classified as the VII class of ecological state were omitted from this research as an outlier (Line 228). This is done because this occurrence can be attributed to the specific location of that JDS sampling point, where a smaller tributary joins the Danube river.

Comment #4

In line 235, is there any specific reason you have considered for using the classification level proposed by Krtolica et al. [2]?

Authors’ Response:

The main reason for developing a new classification of the Danube river basin as proposed by Krtolica et al. (2021) is the significant differences in classification scales among Danubian countries (Line 219). Therefore, developing a unique classification scheme was a reasonable solution.

Comment #5

With the limited data that you have, can you justify the accuracy of your results?

Authors’ Response:

We thank the reviewer for this constructive comment. It is known that every scientific approach has some kind of limitations. Due to the nature of the collected dataset and by trying to avoid overfitting, a 10-fold Cross-validation method was used. The results of testing and Cross-validation were similar, which represents an accuracy credibility confirmation.

Applying the selected types of ML models to some different (larger) datasets may result in different accuracy values (Line 460). Based on the achieved accuracy, we can only state that the selected methodology is applicable to similar size datasets where similar accuracy values are also expected.

Round 2

Reviewer 1 Report

The paper has been improved in accordance with the reviewer recommendations and now deserves to be published.